# Current Evidence for Biological Biomarkers and Mechanisms Underlying Acute to Chronic Pain Transition across the Pediatric Age Spectrum

**DOI:** 10.3390/jcm12165176

**Published:** 2023-08-09

**Authors:** Irina T. Duff, Kristen N. Krolick, Hana Mohamed Mahmoud, Vidya Chidambaran

**Affiliations:** 1Department of Neurosurgery, Johns Hopkins University, Baltimore, MD 21218, USA; iduff.md@gmail.com; 2Department of Anesthesia, Cincinnati Children’s Hospital, Cincinnati, OH 45242, USA; kristen.krolick@cchmc.org (K.N.K.); hmmahmou@purdue.edu (H.M.M.)

**Keywords:** chronic pain, chronification of pain, molecular markers, biomarkers, mechanisms, pediatric pain, developmental, peripheral sensitization, central sensitization, neuroimaging of pain, genetics and epigenetics of pain, neurophysiological markers, EEG, QST

## Abstract

Chronic pain is highly prevalent in the pediatric population. Many factors are involved in the transition from acute to chronic pain. Currently, there are conceptual models proposed, but they lack a mechanistically sound integrated theory considering the stages of child development. Objective biomarkers are critically needed for the diagnosis, risk stratification, and prognosis of the pathological stages of pain chronification. In this article, we summarize the current evidence on mechanisms and biomarkers of acute to chronic pain transitions in infants and children through the developmental lens. The goal is to identify gaps and outline future directions for basic and clinical research toward a developmentally informed theory of pain chronification in the pediatric population. At the outset, the importance of objective biomarkers for chronification of pain in children is outlined, followed by a summary of the current evidence on the mechanisms of acute to chronic pain transition in adults, in order to contrast with the developmental mechanisms of pain chronification in the pediatric population. Evidence is presented to show that chronic pain may have its origin from insults early in life, which prime the child for the development of chronic pain in later life. Furthermore, available genetic, epigenetic, psychophysical, electrophysiological, neuroimaging, neuroimmune, and sex mechanisms are described in infants and older children. In conclusion, future directions are discussed with a focus on research gaps, translational and clinical implications. Utilization of developmental mechanisms framework to inform clinical decision-making and strategies for prevention and management of acute to chronic pain transitions in children, is highlighted.

## 1. Introduction

Acute pain in children (postsurgical, inflammatory, posttraumatic pain or due to other pathologies) is typically protective, subsiding within 14 days, depending on the extent of injury. However, acute to chronic pain transitions are increasingly being described in literature. Chronic pain as a continuum that develops earlier in life is suggested by the evidence that 17% of adults with chronic pain report onset of pain in childhood [1]. In fact, about 20–40% of acute pediatric pain has the potential to transition into chronic pain in children and adolescents [2,3,4,5,6]. Studying chronic pain transitions in the pediatric age group is of utmost importance because (a) children experiencing chronic pain have a ≈3- to 6-fold higher risk of developing chronic pain and disability in adulthood, such that 40–60% of children with chronic pain continue to experience pain as adults [7]; (b) chronic pain in children has greater socioeconomic consequences with a high financial burden [8], stemming from psychosocial and physical disabilities [9,10], impacting quality of life and development of children negatively; and (c) maturational development of the pain connectome is a dynamic process [11], emphasizing neurodevelopmental differences between children, adolescents, and adults’ nociception [12].

In fact, secondary chronic pain such as chronic post-surgical pain has been recognized as an independent entity by being included as a diagnosis under the 11th version of the International Classification of Diseases (ICD-11) [13]. However, the need remains for pediatric specifications for primary pain diagnoses in the ICD-11 [14]. Chronic pain is known to be a complex experience with biological, psychological, and social components. While psychosocial aspects of pain chronification have received much consideration, the biological biomarkers and mechanisms in the pediatric age group have received less attention in the literature, although they are very interrelated. The biological components of pain include sensory and autonomic components of the peripheral and the central nervous system. Neuroendocrine, immune, and genomic mechanisms also participate in injury-induced pain response and chronification of acute pain. This article is stratified by age, and sex as a biological variable has been included in relevant sections.

Currently, there are conceptual models proposed for pain chronification in the pediatric population [15,16], but there exists a need for an integrated theory explaining the mechanisms of acute to chronic pediatric pain transition, factoring in the stages of child development. For example, the narrative for mechanisms of chronic pain development and maintenance have shifted from nociceptive to mesolimbic circuitry in the brain [11,17]. Thus, clinical practice would be better guided by objective biomarkers with the potential to capture the pathological stages of pain chronification. In this article, we review current evidence on mechanisms and biomarkers of acute to chronic pain transitions in infants, children, and adolescents. The available evidence is reviewed in a way that clinicians can integrate the evidence into their daily clinical practice. In addition, this state-of-the-art review article also serves researchers to build upon the best evidence for designing and developing future innovative trials and implementation studies rooted in regarding the biology of chronic pain. Our goal is to help identify gaps and outline future directions for basic and clinical research towards a developmentally informed theory of pain chronification in the pediatric population. We start by outlining the importance of objective biomarkers for chronification of pain in children. Next, we summarize current evidence on the mechanisms of acute to chronic pain transition in adults to draw the contrast with the current evidence on biomarkers of pain chronification in infants, children, and adolescents. Finally, we present a detailed summary of the multifaceted evidence informing developmentally informed mechanisms for transitioning of acute pediatric pain into a chronic state. We conclude our review by discussing future directions, gaps, and clinical implications.

## 2. State-of-the-Art Review

### 2.1. Importance of Objective Biomarkers

In 2006, the Initiative on Methods, Measurement, and Pain Assessment in Clinical Trials (PedIMMPACT) identified eight core outcome domains for clinical trials of pain interventions in children, which included pain intensity, physical functioning, symptoms/adverse events, global satisfaction with treatment, emotional functioning, role functioning, sleep, and economic factors [18]. In 2021, this core set of outcome measures was updated, with biomarkers identified as an emerging priority [19]. Biomarkers are objective measures of biological or pathological processes, or a pharmacological response to therapeutic intervention [20]. A joint FDA-NIH working group for developing biomarker endpoints and other tools identified seven distinct biomarker categories that could be applied across all areas of biological research [20]. Three categories of biomarkers that could be of special importance in chronic pain development in children are predictive, prognostic, and susceptibility/risk biomarkers. Susceptibility/risk and prognostic biomarkers would identify the likelihood (risk stratification) and progression of chronic pain development in children, respectively. Examples are genetic tests, neurophysiological measures, and quantitative sensory tests. Predictive biomarkers would help clinicians to identify patients with a higher chance of responding either favorably or unfavorably to pain treatment. An example is microRNA, *miR-548d*, levels for response to intravenous ketamine in complex regional pain syndrome [21].

Biomarkers for pediatric acute to chronic pain transition are still in the discovery stage. Examples of genetic biomarkers are gene expression changes, single-nucleotide polymorphisms (SNPs), patterns of open or closed chromatin, or polygenic risk scores (PRS). For example, PRS from a panel of 20 variants has been proposed for chronic postsurgical pain (CPSP) in children undergoing spine fusion [5]. Following validation across different cohorts and use of laboratory techniques, genetic biomarkers could be useful for personalizing pain management. Due to use of selective cases and controls for genome-wide association studies (GWAS) [22], PRS score use may not be generalizable across all patient populations. Moreover, pediatric populations and people of non-European descent are not well represented in research studies or within large databases such as the UK Biobank [23], making large-scale studies difficult. In addition, epigenetic changes, such as DNA methylation, can arise from environmental impacts and be passed on mitotically [24], leading to long-term alterations in the regulation of genes with long-term impacts on disease phenotypes [25]. For this reason, the study of DNA methylation or other epigenetic processes in blood for use as biomarkers of acute to chronic pain transition is relevant. For example, DNA methylation levels of 5′-cytosine-phosphate-guanine (CpG) sites in major stress genes in patients aged 18–65 years-old revealed distinct methylation differences predictive of chronic pain symptoms 6 months after injury [26]. In general, pediatric -omics big data are not currently available to conduct large-scale research in pediatric phenotypes. One way to circumvent such handicaps may be the use of systems biology approaches to understand functional pathways associated with CPSP, leveraging evidence from already available animal and human literature [27].

Altered functional connectivity has been identified as a neuroimaging biomarker of chronic pain after acute brain injury in adults - reduced negative functional connectivity of the nucleus accumbens (NAc) with a primary motor cortex region was reported in those who developed chronic pain [28]. Adult patient-specific immune biomarkers, assessed by mass cytometry for single-type specific intracellular signaling molecules, were also predictive of patients’ speed of recovery from surgical pain [29]. As these previous examples were performed in adults, pediatric-population-specific biomarkers are needed. In later sections, we discuss any evidence available from the pediatric literature. It is important to recognize that chronic pain is a complex disease, and utilization of several distinct biomarkers together, such as neuroimaging combined with PRS, may better represent the variable etiology and sub-phenotypes of chronic pain.

### 2.2. Developmental Mechanisms of Pain Chronification

Peripheral and central mechanisms of chronification of pain in adults have been described previously [30,31]. A diagrammatic presentation of these mechanisms is presented in Figure 1. While we acknowledge the pediatric literature remains scarce compared to adults, with gaps in understanding, we present a visual basis of known mechanisms of chronification of pain with pediatric developmental context (from animal/human studies) denoted in blue font in Figure 1, overlaid on the known mechanisms in adults (black lettered headings). The evidence from pediatric studies may be mixed or even contradictory, depending on context, but a detailed description of such nuances is beyond the scope of this review. For example, the direction of connectivity between specific brain regions within the default mode network is different within the same pain condition (for example: lower resting-state connectivity between the posterior cingulate and insula, but greater functional connectivity between the thalamus and posterior cingulate in children with irritable bowel syndrome) or opposite between pain conditions (for example, connectivity was higher, not lower, in complex regional pain syndrome) [32].

Also, the bulk of the literature on developmental mechanisms are from animal studies. The reader is referred to a recent review published by our team [37] for a comprehensive understanding of animal models that currently exist to study acute to chronic pain transition in the pediatric population and the limitations of such models. Briefly, the following rodent models exist for studying pain: local or systemic injections of inflammatory agents into hindpaw or intraperitoneally to model local and systemic inflammation respectively (e.g., see [38,39,40]); transient compression of cervical nerve [41] and spared nerve injury [42,43] models to study neuropathic pain; neonatal manipulations such as repetitive needle pricks [44], injections [45,46], and nonpainful tactile stimulation [47] with or without maternal separation to model and study the stressors neonates experience in the NICU; and hindpaw incision [48,49], which can be used to model pain after surgery [50]. However, as mentioned in our last review, few of these models have been used to study acute to chronic pain transition, with even fewer modeling pediatric age ranges. It is promising though that for adult age ranges, similar genetic and epigenetic findings between humans who experience chronic pain and rodent models are beginning to be found [37].

Animal studies have shown us that the consequences of early life injury vary depending on the type of injury, the sensory modality, and the timing of injury. For instance, repetitive needle pricking of the paw during the first week after birth can result in heat hypersensitivity several weeks later [51]. However, neonatal hindpaw inflammation has a significant impact on behavioral responses and dorsal horn cell activity during a subsequent inflammatory challenge in adulthood [52], but does not induce prolonged heat or mechanical hypersensitivity beyond the first week [53]. Thus, hindpaw inflammation leads to a generalized and slowly developing reduction in baseline sensitivity throughout the body in response to mechanical and thermal stimuli with an increased chance for inflammation later in life. Early life injury can also have the opposite effect. Repetitive formalin injections into neonatal paws can result in generalized heat hypoalgesia in adulthood [54]. These early onset inflammatory hyperalgesia and later-onset baseline hypoalgesia occur only if the original inflammatory stimulus is applied within the first 10 days of life, and both responses persist into adulthood. Long-term hypoalgesia affecting the entire body likely arises from changes in stress response, as exposure to stress during the perinatal period is known to influence nociceptive behavior in adulthood [55]. This adaptation could be seen as a useful response to early trauma. Any long-term sensitization occurring at the segmental level may remain masked and require a strong stimulus, such as second inflammation, to uncover it. In contrast, chemical or mechanical irritation of the colon in rats aged P8–21 leads to persistent visceral hypersensitivity in adulthood [56]. Neuropathic pain development after early life nerve injury presents a different picture—while partial peripheral nerve damage in adult rodents leads to significant and prolonged neuropathic pain behavior characterized by marked allodynia, this does not occur in rat pups up to P21 [34]. Tight ligation of the fifth and sixth lumbar spinal nerves during the first two postnatal weeks produces only transient mechanical allodynia [35], and no changes in sensitivity occur in the spared nerve injury and chronic constriction injury models until P28 [34].

There are several explanations for these findings, including one that it takes time for ascending and descending pathways to form (Figure 1). Also, infant nerve injury triggers an ani-inflammatory immune response in the dorsal horn of the spinal cord (increased interleukin (IL)-4 and IL-10), which is protective. However, after adolescence (post-natal day 25–30), the immune profile of the dorsal horn switches to producing tumor necrosis factor (TNF) and BDNF, pro-inflammatory factors associated with late onset neuropathic pain behavior [36] (Figure 1). Thus, while the neonatal brain and nervous system protect from developing neuropathic pain at this period of time, they still prime it for reactivation later in life if a second injury takes place [36]. Neonatal rats also experience much higher concentrations of growth factors (Figure 1) released by tissue injuries compared to adults, which can influence the development of peripheral nociceptors in various ways [33]. Thus, nerve growth factor (NGF) signaling through trkA receptors leads to enhanced glutamatergic transmission after neonatal incision during a critical time window (first post-natal week) characterized by higher spontaneous signaling and highly sensitive somatosensory circuits, unlike in adults [57,58,59]. Similarly, in the first 2 weeks after birth, neonatal incision increases calcium-permeable AMPA receptors and other ion channel signaling, leading to spike timing-dependent long-term potentiation of responses [60]. Inhibitory synaptic signaling mediated by GABA or glycine is thus weaker at early ages in the spinal cord dorsal horn [61].

Normal maturation of nociceptive circuits requires input from tactile receptors, which guide the nociceptive synaptic organization during the critical stage of development mentioned above [62]. As discussed, intrinsically driven spontaneous activity is prevalent in developing nociceptive circuits which makes them particularly susceptible to permanent reorganization (pruning, altered synaptic connectivity), influenced by altered neuroimmune/endocrine functions in response to early exposure to nociceptive stimuli [63,64]. While the central mechanisms behind the long-term changes in pain behavior are not fully understood, potential mechanisms include alterations in synaptic connectivity and signaling within postnatal nociceptive pathways, as well as shifts in the balance between inhibition and excitation. Descending modulation of spinal nociceptive signaling is immature early in life (Figure 1), and importantly, it favors greater nociceptive transmission [65]. In fact, adult modulatory responses are also modified by early exposure to nociceptive stimuli. Also, in rats, the first postnatal week represented a critical period as incision during that stage (P3-6) increases hyperalgesia following repeat surgery two weeks later. Importantly, use of repeat nerve blocks decreased this hyperalgesic response [66]. This has potential clinical implications for the use of regional analgesia for neonatal surgery.

Furthermore, use of functional magnetic resonance imaging (fMRI) revealed differences in infant cerebral processing of pain perception compared to adults [67]. Interestingly, hyperconnectivity of brain regions seen in chronic pain in adults mirror those involved in neuro-affective disorders in childhood [68,69]. Thus, it has been postulated that “speed-up processes” of neural development through childhood involving increased myelin production and selective pruning might facilitate increased early subcortical–cortical connectivity after early life injury, stress, and pain [32,70]. In addition, pain connectomes are more localized (Figure 1) at younger ages [71], defined more by anatomical proximity, which transition into more specialized neural networks with a more distributed architecture by young adulthood. There may also be a role for “microglial priming” and neuro–glial interactions in the spinal cord and brain in chronic, persistent pain. Microglia in the brain monitor the interstitial fluid and in the presence of threat, resulting in exaggerated neuroinflammatory responses with loss of gray matter (GM) and synaptic pruning. These changes may also have an epigenetic basis. In this context, “neonatal nociceptive priming” is a term coined for increased vulnerability to pain sensitivity to injury in later life, due to early life injury. Neuroimmune interactions related to growth hormone signaling have been elucidated in animal models [72]. There is evidence from animal studies for long-lasting epigenetic remodeling of macrophages that might influence pain memory. For example, the p75 neurotrophic factor was found to regulate inflammatory profile and responses in rodent models and may be a mechanism involved in pain chronification in infants [73].

Thus, neurobiological mechanisms underlying pain chronification may be different in children and adolescents compared with adults. Future studies at the intersection of these different mechanistic processes are needed to quantify the pain dose response, critical development periods, and factors involved in the chronification of pediatric pain [74].

### 2.3. Acute to Chronic Pain Transitions in Infants

#### 2.3.1. Does Neonatal Pain Lead to Chronic Pain Later in Life?

As discussed in the previous section, studies in animals suggest neurotoxic effects of early exposure to pain. An important question remains whether the evidence in animal studies translates to humans. Preterm newborns are known to undergo many potentially painful procedures, including heel sticks, tracheal suctioning, surgeries, etc. On average, it is estimated they are subjected to about 10 daily stress/pain events [75]. In 2005, authors of a cross-sectional study in 164 infants evaluated pain, opioid use, and norepinephrine levels in a subsequent surgery among infants who had major surgery in their first three months of life and compared them to controls. The authors found that, compared to controls, surgery in early infancy led to higher pain as well as higher norepinephrine and opioid requirements when subsequent surgery involved the same dermatomal levels of prior tissue damage [76]. This supports spinal mechanisms of sensitization that could be potentially decreased by use of regional analgesia, as mentioned previously [66]. A stunning statistic revealed by another study evaluating long-term neurobiological, neuropsychological, and sensory development effects of prematurity, procedural pain, and opioids in early neonatal life was that up to 68.4% of children who spend time in the neonatal intensive care unit experienced pain over the 3 months before their visit and 15.8% had chronic pain by ten years of age [77].

An important difference between animal nociceptive studies and neonatal experience is that neonates in the ICU are exposed to not only pain, but also opioids, anesthetics, and other drugs [75,78]. Animal studies on exposure to opioids in early life show opioids were neuroprotective if administrated in the presence of pain and in specific situations [79,80]. A recent meta-analysis showed that a higher number of neonatal pain events, but not opioid administration, in rodents was associated with increased neuronal cell death, increased anxiety, and depressant-like behavior [81]. However, studies in humans show mixed results, which are elucidated in an excellent narrative review [82]. Of note, a longitudinal study found no major effects of neonatal pain nor opioid or anesthetic exposure in the early life in children and young adults (8–19 years of age), using thermal detection, pain thresholds, and high-resolution structural and task-based fMRI during pain. They reported potential neuropsychological effects in the groups with the highest opioid exposure [83], which needs further follow up. Similarly, the study mentioned in the prior paragraph also did not find associations of morphine administration during neonatal life with neurocognitive performance or thermal sensitivity later in childhood [77]. While this is heartening, the authors discuss limitations of sample size and selection bias in their study. In the following sections, we present mechanistic evidence from human studies investigating early pain and stress.

#### 2.3.2. Mechanistic Evidence and Biomarkers for Pain Chronification in INFANTS


**(a) Neuroimaging evidence**


Neuroplasticity plays a crucial role in the formation of neural circuits and the development of GM, white matter (WM), and RS-FC throughout developmental stages, influencing pain perception [84,85,86]. Factors such as pain-related stressors, painful procedures, and exposure to morphine in the NICU are associated with lower global brain volumes and reduced GM throughout the brain during childhood [77]. Smaller brain volumes in general are correlated with lower gestational age, a higher number of painful procedures in the first 14 days of life, and higher exposure to morphine in the first 28 days of life [77]. Abnormalities in the microstructure of white matter are linked to a higher number of invasive procedures by the age of 7 and lower cognitive function [87]. The authors of one study compared cerebral pain response in children and adolescents (11–16 years) using fMRI between groups with experience in a NICU after preterm (≤31 weeks gestational age) and full-term birth (≥37 weeks gestational age) with full term control children without early hospitalization. Compared to controls, significant activations in the thalamus, anterior cingulate cortex, cerebellum, basal ganglia, and periaquaeductal gray were found in the NICU groups, as well as higher activations in primary somatosensory cortex, anterior cingulate cortex, and insula in preterm infants [88]. Also, the authors of a different, longitudinal study in children born very preterm (24–32 weeks gestational age) found that neonatal-pain-related stress was associated with a thinner cortex in multiple (21/66) brain regions at school age (mean 7.9 years), independent of other neonatal risk factors [89]. These findings suggest that the developing brain is both adaptable and susceptible, and changes in pain processing circuits in response to nociceptive stimuli can result in long-term anatomical and functional alterations that contribute to lifelong chronic pain. Structural MRI scans and RS-FC functional brain imaging can reveal signs of neonatal injury, surgery, or inflammation, which could serve as biomarkers for preventive medical treatment of postsurgical/posttraumatic pain in adolescence or adulthood, depending on the timing of secondary insult.


**(b) Electrophysiological evidence**


A wide range of electrophysiological methods are employed to study infant pain processing and the effects of noxious stimulation in both healthy full-term infants and premature infants undergoing multiple procedures in the NICU. These methods include electroencephalography (EEG), magnetoencephalography (MEG), and near-infrared spectroscopy.

Infants who experience stress, multiple medical procedures, or illness exhibit different responses to noxious stimuli. Slater et al. [90] used EEG to determine that noxious-evoked potentials following medically necessary heel lances were larger in ex-premature infants at term-corrected age who had experienced painful procedures in neonatal units, compared to age-matched term-born infants. Jones et al. [91] reported that stress, as measured by salivary cortisol levels, increases noxious-evoked brain activity without a proportional increase in behavioral response. This finding aligns with evidence in adults that stress enhances pain sensitivity [92]. The authors postulated that this disconnect with pain behavior may imply that observation of pain behavior may not be a reliable indicator of pain stress in neonates. Ozawa et al. [93] also demonstrated that prior pain disrupts the relationship between cortical and behavioral measures of pain, emphasizing the need to consider previous experiences when assessing neonatal pain. Prior pain experience from heel lances in the first 24–36 h after birth was reported to elicit pain anticipation and heightened behavioral pain responses to venipuncture in both premature [94] and term-born infants [95]. Importantly, negative responses were mitigated by the use of sucrose [96] and skin-to-skin care [97] during painful procedures.

Mitigation is very important as effects may persist into childhood and adulthood. Children who have experienced early life pain exhibit increased cerebral responses to pain [88] and pain catastrophizing [98]. Additional evidence was derived from a study measuring functional brain activity using MEG and visual-perceptual abilities in school-age children who were born very prematurely (<32 weeks). They demonstrated alterations in spontaneous cortical oscillatory activity and lower perceptual disabilities correlated with cumulative neonatal pain [99], as well as altered network connectivity across standard theta, alpha, beta, and gamma bands in the middle gyrus [100]. This is in line with the thalamo-cortical dysrhythmia model for chronic pain proposed in adults [101,102]. According to this model, abnormal nociceptive input leads to irregular bursts of theta oscillations in the thalamus. These oscillations are then transmitted to the cerebral cortex, resulting in the disinhibition of neighboring areas. Consequently, gamma frequency oscillations occur in the affected regions, leading to the persistence of ongoing pain.

Research on the chronification of pain in infants using electrophysiological markers is still in its early stages. Of note, abnormal neural synchronization [100], low amygdala volumes [77], and altered sub-cortical connectivity [103] have been shown in neuro-affective disorders such as autism. Questions have been raised regarding the potential association of early life stress/pain and life-long influences on neurobehavioral and neuropsychiatric outcomes [104]. In fact, upon implementation of an opioid protocol for NICU babies, Steinbauer et al. reported that the resulting increased neonatal opiate exposure was a potential risk factor for autism spectrum disorder and withdrawn behavior at preschool age. They recommended vigilant use of opiates [105].


**(c) Genetics And Epigenetics**


To understand how many studies have been performed on potential genetic and epigenetic mechanisms of pain chronification in the human infant population, a scoping search (see Appendix A for search terms) for all evidence of genetic and epigenetic alterations after an infant or neonate is subjugated to a painful event was performed. Twelve human studies (see below), along with two reviews [106,107], were returned. While none of the infant epigenetic and genetic studies quantified pain again later in life to truly represent acute to chronic pain transition, three of the infant studies followed the effect of painful procedures through to 4 years [108], 7 years [109], and 8 years [110] after skin-breaks in the NICU (Figure 2a).

Six [108,114,115,116,117,118] out of the twelve human infant studies were by the same team and conducted in Italy between the years of 2015 and 2021. For pre-term infants exposed to high-level, but not low-level, pain-related stress during the NICU stay, as quantified by number of skin break procedures, the methylation of two CpG sites within the *SLC6A4* promotor region increased from birth to discharge [114]. Later, the face-to-face still face paradigm was used as a measure of the mothers’ levels of maternal sensitivity. In pre-term infants, and full-term infants of less-sensitive but not sensitive mothers, increased *SLC6A4* methylation at discharge was associated with a higher negative emotionality (stress-response) at 3 months of age [115]. Thus, it was indicated that in full-term but not pre-term infants, maternal sensitivity served as a protective factor against *SLC6A4* epigenetic variations [115]. A limitation of this study was that pain could not be analyzed as a causal factor in pre-term infants, and instead the authors hypothesized that premature separation of mother–infant contact in the NICU could be the cause [115]. In another study, maternal sensitivity as a protective factor was corroborated—low levels of maternal touch intensified the DNA methylation at CpG sites within the *SLC6A4* promotor that were altered by NICU stay [118]. In another study, the authors followed pre-term infants until 4.5 months of age and analyzed their DNA methylation [108]. Pre-term 4.5-year-olds displayed greater anger in response to an emotional stressor compared to full-term 4.5-year-olds. The methylation of two sites, labeled CpG5 and CpG9, was increased in pre-term compared to full-term infants. CpG5 and CpG9 at discharge were significantly associated with increased anger display [115]. Lastly, this group also conducted two studies on pain exposure, as performed in the NICU on very pre-term infants, and telomere length erosion [116,117]. The authors found that preterm infants who experienced high levels of skin-breaking procedures had decreased telomere length from birth to discharge, while preterm infants exposed to low-levels of skin breaking procedures had no significant difference in telomere length [117], and that this decreased telomere length was predictive of a reduced salivary cortisol response to a stressor (still face paradigm) [116].

Three out of the twelve human infant studies were performed by a group located in Canada [109,110,119]. In 2013, they found reduced hair cortisol levels in 7-year-old children born pre-term compared to full-term [119]. Furthermore, the lower cortisol was associated with greater neonatal pain when in the presence of the minor allele, *NFKBIA* rs2233409, in boys but not girls. In 2014, they measured *SLC6A* promoter methylation for children born very pre-term compared to full-term at 7 years of age [109]. The authors found that at 7 years of age, the children born very pre-term had significantly increased child behavioral problems, as measured by the Child Behavior Checklist questionnaire, and they had significantly increased methylation in 7/10 CpG sites in *SLC6A* compared to those children who were born full-term [109]. After correcting for clinical confounders, neonatal pain and the presence of the *COMT* Met/Met genotype were found to be associated with *SLC6A* methylation [109]. In 2019, they found that in the presence of the minor allele *COMT* 158 MetMet, greater neonatal invasive procedures predicted smaller right hippocampal volumes in ≈8-year-old ex-very-pre-term children, and in the presence of the *BDNF* 66Met allele, a greater number of surgeries predicted smaller right hippocampal subregional volumes in ≈8-year-old children born very pre-term [110].

Out of the remaining studies, two were conducted in Pennsylvania, USA, in 2013 [120] and 2018 [121], and one was conducted in Italy in 2020 [122]. In the 2013 study, 16 out of 31 infants undergoing elective surgery were carriers of the MOR 118G allele and exhibited higher basal skin conductance, a measure of pain, compared to non-carriers [120]. In the 2018 study, no difference in levels of methylation in the first exon of *OPRM1* was found; however, the authors pointed out that this was a preliminary study mainly to obtain data on patient retention, and low sample size (n = 12 NICU-admitted infants) could not be ruled out as the reason for why no differences were found [121]. In the 2020 study, in which ≈1000 neonates were enrolled, infants who were homozygous carriers of the G allele of rs1799971 in *OPRM1* had higher pain scores in response to heel lance after dextrose administration [122].

In conclusion, early life painful events were associated with increased methylation in *SLC6A4* promoter CpG sites (up to 7 years of age), decreased telomere length, and adverse behaviors. Variants in *COMT* and *BDNF* interact with *SCL6A4* DNAm changes to influence outcomes. *OPRM1* and *NFKBIA* variants also potentially influence the stress/pain response to early life pain. The interesting sex difference for *NFKBIA* variant association needs follow up. These findings offer potential as biomarkers to identify neonates at risk for adverse long-term outcomes from early life stress.


**(d) Neuroendocrine evidence**


In utero and perinatal stress might have long-lasting effects on pain processing later in life. The mechanisms underlying the influence of early life stress on pain pathways are mediated by the hypothalamic–pituitary axis (HPA). Prolonged exposure to stress hormones prenatally leads to decreased glucocorticoid receptors in the hippocampus, poorer glucocorticoid feedback sensitivity, and heightened glucocorticoid production in childhood, which manifests as increased anxiety several years later [123]. Neonatal maternal separation, which manifests as stress, has been shown to be associated with reduced sensitivity to noxious heat stimuli [124], as well as visceral hypersensitivity in adult rats [125]. Insufficient nesting material during early life also prolongs muscle hyperalgesia following prostaglandin administration in adulthood, increases the excitability of mature nociceptors innervating the muscle [126], and elevates plasma levels of the pro-inflammatory cytokine interleukin-6 in adulthood [127].

In adults, the HPA axis can directly impact the neurophysiological mechanisms involved in pain perception through brainstem descending pain control pathways [128]. Hypo-responsiveness of the HPA axis in pre-term but not full-term infants has been reported. Specifically, prior neonatal pain exposure (quantified by the number of skin-breaks) was associated with lower plasma cortisol responses to stress in pre-term infants [129]. This corroborates with the evidence presented in above genetics section, where higher number of infant skin-breaks was associated with lower hair cortisol levels in boys 7 years later [119]. Although it is unclear whether the stress response directly influences the development of pain pathways in humans, the immaturity of brainstem descending pain control pathways at birth [65] suggests that increased HPA activity could exert significant modifications. Another important stress axis, the sympathoadrenal system, and its primary mediator, catecholamines, have been implicated in inducing and sustaining the stress-induced maintenance of mechanical hyperalgesia in adults [130], but there is no evidence reported for its involvement in neonatal nociceptive priming.


**(e) Sex differences**


Animal studies reveal age-dependent sexual dimorphism in susceptibility to develop chronic pain [131]. In rats, younger males were found to be less susceptible to TNFα-induced priming compared to older males, and younger females more susceptible compared to older females. This sexual dimorphism was explained by age-related changes in estrogen levels, which is protective [131]. In another study, specifically, female rats were more vulnerable to the long-term consequences of neonatal inflammatory injury [132]. Neonatally injured females exhibited significantly greater hypoalgesia at P60 (adulthood), as well as enhanced inflammatory hyperalgesia following re-injury compared to neonatally injured males and controls [79]. Specifically, in rats, neonatal nociceptive priming susceptibility was dependent on estrogen action on the estrogen receptor, with a neuroprotective effect [131,133,134]. In human infants, the only genetic/epigenetic study analyzing sex differences was in relation to the *NFKBIA* variant. While no association of neonatal pain and cortisol levels was found in girls, in boys, an interaction with the *NFKBIA* variant was found [119]. Sex differences in the HPA axis responsiveness are well established throughout the literature (e.g., see review [135]), and thus future studies on sex differences in early life stress effects on pain pathways in relation to later-life pain chronicity are warranted.

### 2.4. Pain Chronification in Children and Adolescents

In the United States, ≈3.9–6 million children undergo surgery annually [136,137], and injury is the leading cause of death among children older than 1 year. With a reported 14.5–38% (median 20%) incidence of CPSP, the impact on children is high.

### 2.5. Mechanisms and Biomarkers for Pain Chronification in Children and Adolescents


**(a) Neuroimaging evidence**


Structural and functional MRI studies have revealed several brain networks are involved in chronic pain. In children, lower GM and greater RS-FC within the major pain-associated networks were found in several of the chronic pain conditions studied. While functional hubs are mostly confined locally to sensorimotor networks in the early years, with age, they shift to the posterior cingulate cortex and insula [70]. It was recently demonstrated by Jones at et al. that higher pain intensity during adolescent years is potentially associated with desegregation patterns of cerebellum default mode network connectivity [138]. The central executive (memory), salience (expectation response), and default mode networks (emotional processing) are identifiable at an early age and undergo significant change until the age of about 20 years. Baliki et al. conducted a longitudinal study in adult patients with subacute low back pain followed for 1 year for development of chronic pain and compared imaging signals with controls. They found that the functional connectivity between the medial prefrontal cortex and nucleus accumbens in the brain’s emotional learning circuitry was predictive (>80% accuracy) of pain chronification [68]. This connectivity is part of the mesolimbic–prefrontal network, which the authors purported to play a role in reward behavior (with dopaminergic and glutaminergic projections) and addiction potential [139,140]. Interestingly, they found that the prediction was better with longer time lapse between the brain activity measurement and the pain outcomes [141]. The same group also used diffusion tensor imaging and showed that pre-existing brain white matter structural abnormalities were predictive of pain persistence [142]. Readers are referred to an excellent review of a developmental perspective of these networks in pediatrics by Bhatt et al. [32]. Brain imaging research to identify mechanisms of pain chronification in the pediatric age group remains in its infancy. Besides MRI, neural inflammatory markers and neurotransmitters in pain (including creatine, N-acetylaspartate (NAA), myo-inositol, choline, glutamate, glutamine, and gamma-aminobutyric acid (GABA)) may be assessed in vivo by magnetic resonance spectroscopy (MRS) with the advantages of being non-invasive and having no risk of radiation [143]. Neurometabolites detected with MRS have improved the understanding of pathological mechanisms in the pain of fibromyalgia as well as the effectiveness of pharmacologic therapies [144,145]. Another interesting modality is the use of functional near-infrared spectroscopy (fNIRS), a noninvasive optical imaging technique that measures changes in oxygenated and deoxygenated hemoglobin within the brain and has been used in newborns and adults [146]. Using fNIRS, specific changes in the somatosensory cortices are able to be detected (both when patient is or is not undergoing anesthesia), making it a potent and potentially useful technique in evaluating objective pain markers in infants and children [147].


**(b) Electrophysiological evidence**


EEG can be useful in characterizing the neurological processes underlying pediatric chronic pain transitions. EEG frequency, connectivity, and EEG entropy (a measure of EEG information content describing regularity of continuous EEG time series) have been found to characterize chronic pain in adult studies [148]. In adolescents with chronic musculoskeletal pain, increased resting global delta and beta power, changes in EEG spectral power, peak frequency, permutation entropy, and network functional connectivity at specific frequency bands were described during tonic heat and cold stimulations [149]. Similarly, permutation entropy in the theta frequency band was used to classify the presence and absence of chronic pain in female adolescents, showing potential for a point-of-care biomarker to detect pain in the absence of self-report [150]. While it has promise as a non-invasive, low-cost, clinically accessible biomarker [151], it needs to be evaluated further as a marker for chronic pain transitions in children.


**(c) Genetics and Epigenetics**


Figure 2b includes a summary of the major findings from the evidence available for genetic and epigenetic contributions to chronic pain in children and adolescents (see the Appendix A for search terms). Chidambaran et al. (2020) conducted a systematic review and meta-analysis on genetic risk associated with the development of chronic postsurgical pain in humans [152]. Their meta-analysis included 21 full-text articles, of which only one study included pediatric age ranges. They found significant association of variants with chronic post-surgical pain in 26 genes important in neurotransmission, pain signaling, the immune response, neuroactive ligand–receptor interaction, apoptosis signaling, and metabolism and transport [152]. Since not enough genome-wide association studies (GWAS) analyzing chronic postsurgical pain in the pediatric population are available, Chidambaran et al. (2021) used computational biology to develop a PRS predicting the development of chronic pain in the pediatric population. Twenty variants, annotated to seven genes: *ATXN1*, *PRKCA*, *CACNG2*, *DRD2*, *KCNJ3*, *KCNJ6*, and *KCNK3*, comprised the PRS predicting risk of development of chronic pain in 10–18-year-olds [5].

DNA methylation is the main epigenetic mechanism investigated for pain chronification in children. Chidambaran et al. (2017) pyro sequenced 22 CpG sites at the *OPRM1* promoter and found the altered methylation of two CpG sites in adolescents exhibiting chronic pain after spine fusion surgery compared to controls with no chronic pain [4]. In a follow-up analysis using methylation array data, Chidambaran et al. (2019) reported differential DNA methylation of 637 CpG sites in spine-fusion patients (n = 56) aged 10–18 years old who developed chronic postsurgical pain compared to those who did not [111]. These sites were located in 310 genes involved in pathways such as GABA receptor signaling, protein kinase C signaling, dopamine receptor, and cAMP-mediated signaling [111].

There is scarce research investigating the cross-section of gene–epigenetic interactions in the development of chronification of pain in children. In 2021, a pilot study by Chidambaran et al. (2021) analyzed DNA methylation association with SNPs in 10–18-year-olds (average age of 14) undergoing spine fusion surgeries [112]. They found DNA methylation at 127 CpG sites mediated the association of 470 methylation quantitative trait (meQTL) loci with chronic pain [112]. Important CpG meQTL sites were located within 5 genes, namely, *SLC45A3*, *NUCKS1*, *RAB7*, *RAB7L1*, *SLC41A1*, and *PM20D1* [112]. Thus, these genes were deemed to be important in terms of epigenetic regulation risk for developing chronic pain after surgery in pediatric patients [112].

Transcriptomics and other -omics-related investigations of pain chronification in children are also lacking in the literature. The RNA sequencing of peripheral blood in adolescents undergoing pediatric spinal fusion surgeries revealed the increased expression of *HLA-DRB3* in adolescents who experienced chronic postoperative pain compared to those who did not, albeit the authors concluded they may have had a low sample size [113].


**(d) Neuroendocrine evidence**


Stress-associated biomarkers also play a role in the chronification of pediatric pain. Allostatic load [153] is a biological indicator of chronic stress. It is evaluated as a summed clinical index of neuroendocrine, cardiovascular, metabolic [154], immune, and inflammatory markers [155,156,157]. In a study of 61 children and adolescents with chronic pain, over 50% were classified as at high risk for allostatic load, indicating the role of chronic stress in chronic pain [156]. Although stress could be related to the pain condition itself, we also know that allostatic load correlates with adverse childhood events (ACE) and socioeconomic status, which indicates the possibility of stress preceding the development of chronic pain [158,159]. In fact, children with exposure to one or more ACEs had higher rates of chronic pain (8.7%) as compared to those with no reported ACEs (4.8%) [160]. Since ACEs are also associated with post-traumatic stress, which mediates the association of ACEs with mental health problems, it is important to assess ACEs [161,162] in children with chronic pain to inform mitigating strategies.

In patients with a history of early childhood trauma, increased methylation of CpG sites within the *TRPA1* promoter was later associated with increased mechanical pain thresholds in female adult patients with chronic pain [163]; the *FKBP5* variant rs3800373 was associated with right hippocampal volume in adults with musculoskeletal pain [164]; and altered *DRD2* expression in female children and altered *COMT* expression in male children was found [165]. It was speculated that epigenetic changes conferred by the parents’ early adverse experiences were inherited in their children in a sex-dependent manner [165]. In another study conducted in the Netherlands involving 2980 adults, early life stress before the age of 16 was associated with higher musculoskeletal pain scores in adulthood [166]. This provides aa rationale for the investigation of stress-related genomic, and other, mechanisms in pain development (which has a lot in common with certain mental health conditions) along the childhood–adult continuum [167,168].


**(e) Sex differences**


Female sex has been shown to have a higher risk of acute to chronic pain transitions in adults [169,170,171]. Since female adolescents have a higher predilection for chronic pain conditions [172,173], one would hypothesize a higher risk for female sex for pain chronification in adolescents as well. Most studies in children undergoing surgery have not identified sex as a predictor of acute to chronic pain [6,174,175]. Suryakumar et al. did find a higher incidence of females developing chronic postsurgical pain after spine fusion; however, scoliosis has a female preponderance [176], and this might have influenced the results. Genetic interactions with sex were mentioned in previous sections (Section 2.5, (d)). Sex differences based on psychophysical tests are also mentioned in the next paragraph (Section 2.5, (f)). For a more comprehensive review on future priority areas to understand sex and gender differences in chronic pain, the reader is referred to these excellent reviews [177,178]. Further investigations are needed to understand the role of sex, hormonal mechanisms, and gender identity in acute to chronic pain transitions in pediatrics, especially given the pubertal transitions happening through adolescence.


**(f) Psychophysical mechanisms**


Quantitative sensory testing (QST) may reveal mechanisms of chronification from alterations in pain processing. QST provides non-invasive ways to interrogate large fiber function (Aβ), nociceptive small fiber (Aδ, C) function, and the spinothalamic pathways involved in pain chronification. A prospective longitudinal study in children aged 10–17 years conducted QSTs and followed participants for 4 months after acute musculoskeletal pain complaints. The incidence of chronification was 35%. They found that poor conditioned pain modulation using hot and cold thermal tasks was predictive of chronic pain [179], suggesting that impairment in inhibitory pain modulation may predict children’s nonrecovery from acute musculoskeletal pain. Further studies are warranted evaluating mechanisms of central sensitization (using temporal summation) along with other pain paradigms (e.g., pressure pain). The evidence in adults suggest QST may be useful in a mechanism-based classification of pain [180], but there are gaps in our current understanding of QST in pediatric populations and age limitations in its applicability. Of note, QST protocol of the German research network on neuropathic pain (DFNS) encompassing all somatosensory modalities with modification of instructions and pain rating was evaluated for use in 176 children aged 6–12 years. This study by Blankenburg et al. found that QST was feasible for children over 5 years of age [181]. There were differences by body site (face more sensitive than the hand and/or foot), age (children aged 6–8 years were less sensitive to all thermal and mechanical detection stimuli but more sensitive to all pain stimuli than children aged 9–12 years who were similar to adolescents [13–17 years]), and sex (girls were more sensitive to thermal detection and pain stimuli, but not to mechanical detection and pain stimuli). Reference values were shown to differ from adults, but distribution properties (range, variance, and side differences) were similar and plausible for statistical factors. This seminal study demonstrated that developmental changes influenced reference values in children differently than from adults’, while other properties were similar to adults. Clinicians need further evidence-based QST protocols that are feasible and meaningful in clinical settings with high sensitivity and specificity for pain sub-types.

## 3. Future Directions for Research

The field of preclinical pediatric pain research has provided high-quality evidence for the development of maladaptive nociception and the effect of early injury on the development of inflammatory and neuropathic pain [54,84,182,183,184,185]. However, there are gaps in understanding prenatal and early postnatal insults across the continuum of a child’s life [76]. While numerous animal studies suggest nociceptive priming related to stressful/painful experiences in early life and maladaptive pain responses in later life, human studies are not conclusive. Some find that long-term neuropsychological effects were only in those with highest opioid exposure, while others reinforce susceptibility to chronic pain by 10 years of age in 13% of children exposed to early life pain. On one hand, this also points to the need for better predictive preclinical models mimicking real life in NICU, the need for reliable objective measures of pain chronification, and a paucity of mechanism-based validated targets. On the other hand, they suggest the need for continued research to determine dose response of preemptive morphine and other medications targeting mechanistic pathways in attenuating long-term, behavioral impact of neonatal pain and preventing hyperalgesia [79]. While psychophysical tests to determine mechanisms can be used in children as young as 5 years, epigenetic tests and neuroimaging/EEG biomarkers could be relevant in younger children to understand the risk for chronic pain development in those with exposure to early life stressors.

There is also a dire need for multisite longitudinal pediatric studies with inclusion of diverse patient populations that are heterogeneous across multiple pain conditions, thus generating “big data” shared warehouses with well-defined phenotypes and well-characterized mechanisms accessible to all pain researchers. The multidimensional nature of pain emphasizes the need for investigations of multi-modal composite pain biomarker signatures for pain sub-phenotypes [186]. Recent advances in innovative analytical tools in artificial intelligence and machine learning, such as deep learning with neural networks, may allow pattern detection using several parameters from patient electronic databases. Patterns could be leveraged for the automated risk prediction of sub-phenotypes. Quantifying risk and understanding mechanisms could then guide healthcare providers to use tailored preventive strategies, despite the paucity of validated targets. Furthermore, artificial intelligence can be used to discover new drug molecules, and precision gene therapy practices may be utilized. The emphasis on pain biomarkers has been amplified by the National Institute of Health Helping to End Addiction Long-term (HEAL) initiative, the Biomarkers Consortium, and the National Institute of Health Blueprint for Neuroscience Research Grand Challenge of Pain. Currently, there are no clinical biomarkers for pain approved by the FDA. This further emphasizes the need for basic science, translational, and clinical researchers to collaborate on these efforts to develop qualitative and quantitative biomarkers to supplement patient self-reported outcomes for a more comprehensive assessment of treatment responders.

Thus, future research needs to target genetic, epigenetic, immune, electrophysiological and imagining biomarkers in appropriate pain models. For example, antagonists of voltage-gated sodium channel NaV1.7 are under investigation for treatment of certain pain conditions [187]. Given the implications for *KCNS1* channel genes and *PM20D1* in CPSP transition, future studies targeting these mechanisms may reveal new therapeutics [188,189]. In addition, tenazemub, an anti-NGF antibody (phase 3 trials), and inhibitors targeting TRPV1, TRPA1, and TRPM2 receptors (pre-clinical phase), are being investigated for pain conditions but mechanistically could be useful for pain chronification as well [190,191]. For example, treatment with TRP antagonists in mice with acute pancreatitis slowed progress to chronic pain [192]. Research is still needed to identify molecules that can enhance *KCNS1* function. In addition, it is possible that several nutritional, environmental, and psychological factors have shared epigenetic underpinnings with pain sensitization (for example, vitamin D deficiency, socioeconomic status, anxiety sensitivity) [193]. Epigenetic biomarkers can thus serve as prognostic indicators of therapy response [194,195,196,197]. Some examples of epigenetic drugs with successful clinical use for the treatment of hematologic malignancies are DNA methylation transferase (DNMT) inhibitors (5-azacytidine and decitabine). Similarly, inhibitors of DNMT, histone acetyl transferase (HAT), and histone deacetylase (HDAC), enzymes involved in methylation and histone acetylation, as well as CRISPR-mediated methylation or demethylation of specific genes [198], are promising targets to be pursued [199]. Continued exploration of epigenetic drugs is warranted with the means to avoid off-target negative effects. Integration of DNA methylation studies with transcriptomics, proteomics, and chromatin accessibility will allow better inferences regarding druggable targets such as transcription factors. Future research could leverage systems biology and bioinformatics to understand downstream signaling effects of drug targets to enable innovation in a cost-effective manner. Lastly, while evidence of sex–genome interactions in pain chronification have been reported in previous sections, it needs further study in pediatric populations.

## 4. Future Directions for Clinical Practice

To enable the implementation of research informed strategies, clinicians will need to elicit a detailed patient history regarding early life circumstances, previous pain experiences, socioeconomic environmental factors, etc., when children present with acute pain conditions or surgery. We need more validated, easy-to-use psychosocial measures for risk stratification in the clinic, for example, the Pediatric Pain Screening Tool for CPSP [176]. Psychophysical testing tools and standards need to be adapted for implementation at bedside for children. For example, in adults, use of von Frey filaments for temporal summation, pressure pain sensitivity using an algometer or a modified approach using a blood pressure cuff, and Neuropen for mechanical pain detection have been shown to identify sub-phenotypes [200,201]. Knowing baseline QST phenotypes may be able to predict efficacy of pregabalin, lidocaine, oxycodone, and placebo analgesia [202,203]. In addition, point-of-care genetic/epigenetic testing using blood or saliva samples would enable modification of analgesia regimens targeting altered genomic pathways, rather than the current trial-and-error approach [194,196,204]. Finally, we need to consolidate consensus guidelines derived from clinical evidence to support the choice of treatment intervention, based on mechanisms [205,206,207]. Future trials to prevent CPSP should investigate efficacy of interventions based on patient risk and mechanisms. For example, presurgical interventions to address psychosocial risk factors using behavioral interventions [208], functional disability using physical therapy, peripheral sensitization using regional analgesia, and/or central sensitization using NMDA antagonists or calcium channel blockers such as gabapentinoids [209,210,211]. Of note, current literature is inconclusive about the efficacy of pharmacological modalities or regional analgesia in preventing CPSP, mostly due to the heterogeneity of pain conditions, populations, small study sizes, variations in dosage, timing and duration of treatment, and variations in outcome measures [210,211,212,213,214,215]. Future studies with risk stratification and longer-term follow-up for multimodal pain protocols are warranted. In the case of children, family-centered care approaches are needed as impaired parental responses may also reinforce the child’s pain response when parents have been involved with the NICU experience, indicating the need for parental education as well [216]. Importantly, we need policies supporting biomarker development and implementation, improved reimbursement for evidence-backed pain management practices, improved leverage of electronic media for education, assessments and management, and most importantly the support within the electronic medical record infrastructure to enable automated clinical decision support tools based on risk stratification.

Furthermore, the plasticity of the developing nervous system and mechanistic approaches imply reversibility. Thus, there is potential for modulating the processes favorably by use of physiologic interventions such as neuromodulation, virtual reality (VR), and psychological therapies [217]. Neurostimulation is a neuromodulatory method in which electrical impulses are delivered invasively or non-invasively to stimulate peripheral nerves, the spinal cord, or specific brain regions [218,219]. Ilfeld et al. have published extensively on the use of peripheral nerve stimulation as an opioid-sparing technique for postoperative pain [220,221]. However, its benefits for children and prevention of chronification remain to be proven. Central stimulation techniques have been investigated for the management of acute postoperative pain and prevention of chronic migraine, and they hold promise for the prevention of pain chronification [222]. Further work is needed to prove safety/efficacy in children for the prevention of pain chronification [223,224]. In addition, VR, a technology that provides an immersive experience in a simulated and interactive environment using multimodal sensory stimuli inputs, is increasingly being used in pain management [225]. Its application for post-traumatic distress and anxiety, as well as improvement in conditioned pain modulation efficiency, suggest it can influence central sensitization mechanisms and brain modulation of pain with the potential for long-lasting effects [226]. Although the concern may be that need for patient participation may limit its utility to developmentally mature children, VR has been used successfully in children as young as five years of age and children with cerebral palsy [227]. As children’s brains process VR differently and fantasies might be confused with reality, virtual environments may need to be contextualized specifically to enable this distinction [228]. Other non-pharmacological modalities with positive effects on maladaptive pain processing are cognitive behavioral therapy (CBT) and mindfulness-based approaches, which have been used in adolescents. CBT for 4–11 weeks has been shown to restore RS-FC between the insula and somatosensory regions, with long-standing effects on pain outcomes and pain catastrophizing [132,229]. A Cochrane review of the use of psychological therapies for children with recurrent/chronic pain concluded that face-to-face psychological therapies are effective for reducing pain for children with headaches, as well as reducing pain intensity and disability in children with mixed chronic pain conditions [230]. However, the evidence for effects on anxiety and depression, as well as the long-term effects, were inadequate. In addition, integrative care and music therapy are effective treatment paradigms for the treatment of pediatric pain in different contexts [231,232,233], but they are less explored in the context of chronic pain transitions. Thus, the evidence for the effectiveness of most treatment paradigms is mixed [234]. It is likely that mechanistic trials of treatment paradigms stratified by genetic, psychophysical, and psychological risk assessments may improve the ability to discern responders. Table 1 lists the several mechanism-based available and potential pharmacologic and non-pharmacologic therapies in pain chronification [216]. However, early identification, prevention, and management are of utmost importance.

## 5. Conclusions

This state-of-the-art review article describes the evidence for the complex interplay between genetic, epigenetic, chemical, neuronal, and immune factors that interact synergistically to influence the development and maintenance of chronic pain in children. While innate pain perception and responses may be determined by genetic underpinnings, environmental experiences are encoded through functional and structural reorganizational processes in the plastic, developing nervous system to shape memory and behaviors through central and peripheral sensitization mechanisms. These can be detected using several psychophysical, imaging, and genomic tests, and reversed using tailored precision medicine approaches. Critical gaps in research and its translation into clinical practice were identified. These are barriers that need to be surmounted for successful innovation in the promising field of pain chronification in children. The implication for clinicians is that developmental aspects and mechanism-based treatment should be considered while caring for pediatric age groups. In caring for neonates undergoing painful/stressful procedures, there is an imperative to consider soothing therapies (sucrose analgesia, skin-to-skin care), and for neonates undergoing surgeries, the use of regional analgesia, as well as the judicious use of opioids and other analgesics to prevent hyperalgesic priming. This is especially important given that the risk is higher during the critical time of early development. In addition, since pain behavior may not correlate with brain markers during stress in neonates, reliance solely on pain behaviors may lead to undertreatment. Knowing that there is a risk for pain sensitization by 7–10 years of age, it is important to consider the history of early life stress and ACE factors while planning care management for future painful procedures, for behavioral optimization, and to prevent amplified pain responses and sustenance. Although genomic biomarkers for pain chronification are not validated and there are no FDA-approved biomarkers, the use of psychosocial screeners, currently available genetic tests for opioid PK/PD, bedside QST, and portable EEG hold promise to inform clinicians of potential sensitization. This will allow targeted and tailored measures to prevent chronification of pain upon exposure to another acute pain/stress episode by optimizing cerebral responses using non-pharmacological, behavioral, and neuromodulation therapies. Thus, mechanism-guided multimodal pharmacological therapies and regional techniques to provide analgesia and prevent exaggerated responses is especially important as the brain areas involved in pain chronification may also overlap with risk-reward salience areas involved in addiction. Future studies should focus on biomarker-guided risk stratification and the development of personalized non-opioid strategies for pain management. Another emerging concern at the crossroads of pain chronification and mental health stems from the potential overlap of primed pain responses and long-term risk for neurobehavioral and neuropsychiatric outcomes after exposure to early life stress. Future efforts to further understand this risk and mitigate it are warranted.

## Figures and Tables

**Figure 1 jcm-12-05176-f001:**
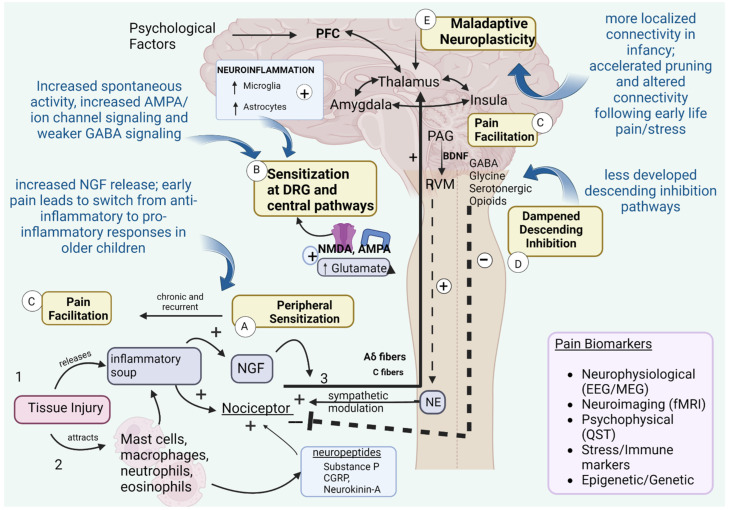
Peripheral and central mechanisms associated with chronification of pain with relevance to the pediatric population (based on animal studies). The main adult mechanisms are mentioned in bold black color denoted by letters, with pediatric-specific features that contribute to pain chronification overlaid in blue-colored text. Figure depicts (1) tissue injury as an acute injury, which then (2) recruits mast cells, macrophages, neutrophils, and eosinophils to the area. Both these cells and tissue injury all contribute to release of the inflammatory soup (proinflammatory cytokines, prostaglandins, histamine, nitric oxide, serotonin, and NGF), which increases NGF. (3) NGF binds to TrkA at peripheral ends of the sensory nerve fibers (C-fiber and A-δ fibers), leading to upregulation and stimulation of Na^+^ channels and causes peripheral sensitization (A). When NGF binds to TrkA on afferent inputs inside the dorsal root ganglion of the spinal cord, it prolongs their stimulation and increases glutamate release onto NMDA and AMPA receptors, resulting in central sensitization at the level of the spinal cord dorsal root ganglion (B), while neuroinflammation due to increased microglia and astrocyte responses leads to central sensitization (B). This is also facilitated by weaker GABA signaling and increased spontaneous firing. Chronic/recurrent stimulation and activation of PAG results in release of BDNF into the RVM, and BDNF-TrkB signaling leads to pain facilitation (C). Decreases in GABA, glycine, serotonergic, and opioid responses, leading to decreased inhibitory sympathetic output, contributing to pain sensitization, also known as decreased descending inhibition (D). Lastly, maladaptive neuroplasticity (E) takes place, contributing to central sensitization of pain. Pediatric developmental context from available studies is shown in blue: For example, increased growth factor release by tissue injury in neonatal rats [33], nerve injury in neonatal rats produces more anti-inflammatory responses and descending inhibition is not developed [34,35,36], earlier synaptic pruning, altered pain connectivity and myelination [32]. Abbreviations: α-amino-3-hydroxy-5-methylisoxazole-4-propionic acid (AMPA), brain-derived neurotrophic factor (BDNF), calcitonin gene-related peptide (CGRP), dorsal root ganglion (DRG), electroencephalography (EEG), functional magnetic resonance imaging (fMRI), γ-aminobutyric acid (GABA), magnetoencephalography (MEG), nerve growth factor (NGF), N-methyl-D-aspartic acid (NMDA), norepinephrine (NE), periaqueductal gray (PAG), prefrontal cortex (PFC), quantitative sensory testing (QST), rostral ventromedial medulla (RVM), tropomyosin-related kinase A (TrkA). Created with BioRender.com.

**Figure 2 jcm-12-05176-f002:**
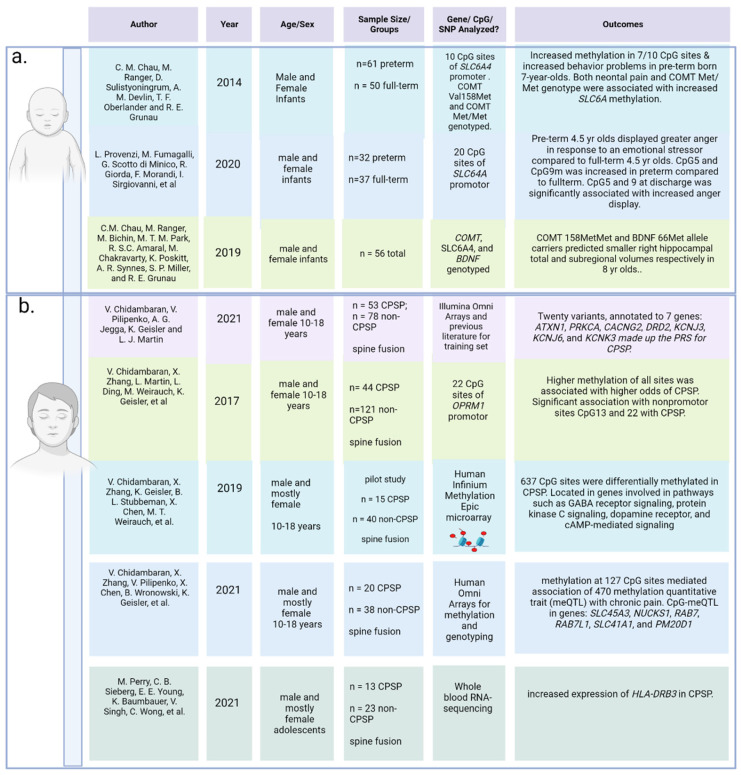
Genetic and epigenetic evidence of acute to chronic pain transition in the pediatric population. Summary of the genetic and epigenetic evidence found for acute to chronic pain transition in the human pediatric population. (**a**) Evidence from infant studies [108,109,110], and (**b**) from child/adolescent studies [4,5,111,112,113]. Abbreviations: *Ataxin 1* (*ATXN1*); brain-derived neurotrophic factor (BDNF); chronic postsurgical pain (CPSP); *calcium voltage-gated channel auxiliary subunit gamma 2* (*CACNG2*); *catechol-O-methyltransferase* (*COMT*); 5′-cytosine-phosphate-guanine (CpG); *dopamine receptor D2* (*DRD2*); *major histocompatibility complex*, *class II*, *DR beta 3* (*HLA-DRB3*); methylation quantitative trait loci (meQTL); *nuclear casein kinase and cyclin-dependent kinase substrate 1* (*NUCKS1*); *opioid receptor mu 1* (*OPRM1*); *peptidase M20 domain-containing 1* (*PM20D1*); polygenic risk score (PRS); *potassium inwardly rectifying channel subfamily J member 3* (*KCNJ3*); *potassium inwardly rectifying channel subfamily J member 6* (*KCNJ6*); *potassium two pore domain channel subfamily K member 3* (*KCNK3*); *protein kinase C alpha* (*PRKCA*); *RAB7*, *member RAS oncogene family* (*RAB7*); *RAB7*, *member RAS oncogene family-like 1* (*RAB7L1*); *solute carrier family 6 member 1* (*SLC6A*); *solute carrier family 41 member 1* (*SCL41A1*); *solute carrier family 45 member 3* (*SLC45A3*). Figure created with BioRender, modified from a template by Ruslan Medzhitov (Creator), Akiko Iwasaki, and Wendy Jiang (See the Appendix A for details on the search strategy).

**Table 1 jcm-12-05176-t001:** Examples for mechanism-based (available and potential) approaches to the management of children at risk for pediatric chronic pain transitions.

Targeted Mechanisms	Pharmacological Interventions * (Examples)	Non-Pharmacological Interventions (Examples) Neuromodulation and Behavioral Therapy	
Peripheral sensitization	Non-steroidal anti-inflammatory drugs such as ibuprofen, celecoxib	Peripheral nerve stimulation (PNS) Vagal nerve stimulation	
	Regional analgesia techniques including peripheral nerve blocks and neuraxial analgesia		
	Capsaicin cream		
	Topical application/infiltration of local anesthetics such as lidocaine		
	In phases of trials: anti-NGF antibody (phase 3); TrkA receptor antagonist (phase 2)		
	Potential: TNF blockers such as adalimumab		
Central sensitization/pain facilitation	Agonists at α2δ (alpha-2-delta) subunit of presynaptic voltage-sensitive Ca^2+^ channels: gabapentin and pregabalin	Spinal cord stimulation (SCS) Transcutaneous electrical nerve stimulation (TENS)	
	NMDA antagonists (ketamine, methadone, dextromethorphan)		
Descending pain inhibition	Tricyclic antidepressants (amitriptyline)		
	Serotonin norepinephrine reuptake inhibitors (duloxetine)		
	Clonidine/dexmedetomidine		
Cortical modulation of pain	Anxiety medications such as benzodiazepines	Invasive neurostimulation: deep brain stimulation (DBS)	Virtual reality immersive therapy; distraction cognitive behavioral approaches; mindfulness integrative care relaxation, music, etc.
		Noninvasive brain stimulation: transcranial magnetic stimulation (TMS) and transcranial direct current stimulation (tDCS)	

* Opioids can potentially influence all mechanisms.

## Data Availability

Not applicable.

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
