# Peer review of "Current Evidence for Biological Biomarkers and Mechanisms Underlying Acute to Chronic Pain Transition across the Pediatric Age Spectrum"

_jcm, 2023, doi:10.3390/jcm12165176_

Round 1

Reviewer 1 Report

The article titles "Current evidence for biological biomarkers and mechanisms underlying acute to chronic pain transition across the pediatric age spectrum." by Duff et al. provides a review of current evidence of biomarkers we have on acute to chronic pain transitions in pediatric populations. Overall, its a well written article with some comprehensive evidence pointing to the prevalence of acute to chronic pain transitions in pediatric populations, and its impact on adulthood pain. I only have a few comments.

1. Given the background of the authors, it is reasonable that they have more information to provide on the genetics/epigenetics portion of the paper, while the electrophysiology/metabolic studies are skimmed through. For example, papers such as PMID: 32954244 that describe neurologic pain signature (nps) in pediatric population have not been included. It would be more comprehensive if authors include some of the works by Wager et al. (and similar) that are relevant to pediatric population.

2. Can the authors include information about gender differences and its relevance to pain in pediatric population.

3. genetics/epigenetics portion in the children/adolescent section could benefit from a table.

4. Cancer pain is also a condition that is more prevalent in children. Can the authors include some evidence on mechanisms of cancer pain in children?

Author Response

Re: JCM Manuscript ID: jcm-2517340

Dear Editor,

We are very excited about the quick turn-around time and the constructive feedback received from you and the Reviewers. We have addressed every comment to the best of our abilities given the scope of the article. All text changes made to address the reviewers’ comments are marked in red font in the revised manuscript. Furthermore, a graphical abstract has been added as requested. Other changes include references relevant to the contents of our manuscript and changes to correct any remaining grammatical errors. All the co-authors have contributed and approved this revised manuscript.

Please see below for our point-by-point responses to each Reviewer. We look forward to publishing this manuscript on a subject that we believe is of great importance.

Sincerely,

Irina Duff, Kristen Krolick, Hana Mahmoud, and

Vidya Chidambaran, MD, MS, FASA

Edward E. Lowe, MD Chair for Clinical and Translational Research in Anesthesia Professor of Anesthesia Director of Perioperative pain, Division of Pain Management Department of Anesthesia Cincinnati Children's Hospital

3333 Burnet Ave, MLC2001, Cincinnati, OH 45229

Response to Reviewer 1

[General ranks]

Yes      Can be improved      Must be improved    Not applicable

Does the introduction provide sufficient background and include all relevant references?

(x)        ( )        ( )        ( )

Are all the cited references relevant to the research?

(x)        ( )        ( )        ( )

Is the research design appropriate?

( )        ( )        ( )        (x)

Are the methods adequately described?

( )        ( )        ( )        (x)

Are the results clearly presented?

(x)        ( )        ( )        ( )

Are the conclusions supported by the results?

(x)        ( )        ( )        ( )

[General Comment] The article titles "Current evidence for biological biomarkers and mechanisms underlying acute to chronic pain transition across the pediatric age spectrum." by Duff et al. provides a review of current evidence of biomarkers we have on acute to chronic pain transitions in pediatric populations. Overall, its a well written article with some comprehensive evidence pointing to the prevalence of acute to chronic pain transitions in pediatric populations, and its impact on adulthood pain. I only have a few comments.

Response: Thank you very much for your kind remarks.

[Comment 1] 1. Given the background of the authors, it is reasonable that they have more information to provide on the genetics/epigenetics portion of the paper, while the electrophysiology/metabolic studies are skimmed through. For example, papers such as PMID: 32954244 that describe neurologic pain signature (nps) in pediatric population have not been included. It would be more comprehensive if authors include some of the works by Wager et al. (and similar) that are relevant to pediatric population.

Response: Thank you for noting our backgrounds. Thank you for suggesting the PMID-32954244 reference. It provides an excellent comparison between infant and adult pain processing utilizing a whole brain fMRI recording.  We have bolstered the sections on electrophysiology and non-genetic sections as well. We have now included it in section 2.B developmental mechanisms when setting up the fact that there are pain differences between infants and adults. (P.6, L.252)

[Comment 2] 2. Can the authors include information about gender differences and its relevance to pain in pediatric population.

Response: While ‘gender’ means the sex that one identifies with, ‘sex’ is usually the term reserved in the literature for describing biological sex differences. Since sex differences in pain is beyond the scope of our current review, we agree it is important to include sex as a mechanism for acute to chronic pain transition in pediatrics. As such, we have added “sex differences” sections: L.503-519, P.11-12 in the infant section, and L. 651-665, P. 14 in the child/adolescent section.  Information throughout the text that touch on sex differences (some added, and some were in old submission) appear in L. 60, 442-444, 471, 636, 688, and 758-759. These are in genetics, QST, HPA axis and other areas.

[Comment 3] 3. genetics/epigenetics portion in the children/adolescent section could benefit from a table.

Response: Figure 2 was the combined result of the epigenetic and genetic studies. Since it was created in Biorender, it is exported as a figure and not a table. Labels of “a.” and “b.” have now been added to Figure 2 in order to clarify that the “a.” part is for infants, and the “b.” part is for children/adolescents. For the infant studies, only those studies which showed evidence of acute to chronic pain transition were included in the figure (i.e., only 3/13 infant studies talked about in the infant section 2.C genetics and epigenetics part). Text was clarified and changed to Fig.2a or Fig.2b in infant and child/adolescent sections respectively.

[Comment 4] 4. Cancer pain is also a condition that is more prevalent in children. Can the authors include some evidence on mechanisms of cancer pain in children?

Response:. Cancer pain could technically be labeled as a type of neuropathic pain, but it is such a broad topic that it would be a separate review. It is not within the scope of our current review of acute to chronic pain transitions.

Reviewer 2 Report

I really appreciate the opportunity to review this manuscript entitled “Current evidence for biological biomarkers and mechanisms underlying acute to chronic pain transition across the pediatric 3 age spectrum.” This is important to explore biological biomarkers in chronic pain in this population.  I only remark some issues (most of them in the state of the art) in order to improve the quality of this manuscript.

The abstract is clear but I recommend use the third person for formal writing avoiding “we”. Introduction was well structure and shows the necessity for this research. The aim of the paper is clear at the end of the introduction.  

At the state-of-art section, there are some questions that should be review.  Figure 1 is only about children? Only about adults? Both? It should be detail each part. On the other hand, in the Electrophysiological evidence epigraph, when you are explaining the importance of stress and pain catastrophizing what are the clinical implications of this findings? Are there evidence about management of that? Also, on page 11, when you are exploring the importance of the emotional processing. What are the clinical implications?

In the future directions for clinical practice epigraph, in Table 1, which is very interesting you expose some mechanism for cortical modulation of pain, such us integrative care relaxation, which are the references?

Conclusions were correct but should also include a summary of the clinical implications.

Author Response

Re: JCM Manuscript ID: jcm-2517340

Dear editor,

We are very excited about the quick turn-around time and the constructive feedback received from you and the Reviewers. We have addressed every comment to the best of our abilities given the scope of the article. All text changes made to address the reviewers’ comments are marked in red font in the revised manuscript. Furthermore, a graphical abstract has been added as requested. Other changes include references relevant to the contents of our manuscript and changes to correct any remaining grammatical errors. All the co-authors have contributed and approved this revised manuscript.

Please see below for our point-by-point responses to each Reviewer. We look forward to publishing this manuscript on a subject that we believe is of great importance.

Sincerely,

Irina Duff, Kristen Krolick, Hana Mahmoud, and

Vidya Chidambaran, MD, MS, FASA

Edward E. Lowe, MD Chair for Clinical and Translational Research in Anesthesia Professor of Anesthesia Director of Perioperative pain, Division of Pain Management Department of Anesthesia Cincinnati Children's Hospital

3333 Burnet Ave, MLC2001, Cincinnati, OH 45229

Response to Reviewer 2

[General ranks]

Yes      Can be improved      Must be improved    Not applicable

Does the introduction provide sufficient background and include all relevant references?

(x)        ( )        ( )        ( )

Are all the cited references relevant to the research?

( )        (x)        ( )        ( )

Is the research design appropriate?

( )        (x)        ( )        ( )

Are the methods adequately described?

( )           (x)          ( )           ( )

Are the results clearly presented?

(x)        ( )        ( )        ( )

Are the conclusions supported by the results?

( )           (x)          ( )           ( )

[General comment] I really appreciate the opportunity to review this manuscript entitled “Current evidence for biological biomarkers and mechanisms underlying acute to chronic pain transition across the pediatric 3 age spectrum.” This is important to explore biological biomarkers in chronic pain in this population.  I only remark some issues (most of them in the state of the art) in order to improve the quality of this manuscript.

Response: Thank you very much for your kind remarks. Development of objective biological biomarkers of the acute to chronic pain transition will improve identification of pediatric risk groups and preventive measures for pain chronification.

[Comment 1] The abstract is clear but I recommend use the third person for formal writing avoiding “we”. Introduction was well structure and shows the necessity for this research. The aim of the paper is clear at the end of the introduction. 

Response: Although it is generally acceptable to use the words “we” (a.k.a. first-person point of view) in abstracts, introductions, discussions, and conclusions, we applied your suggestion, and the word “we” is dropped from the abstract, making it much stronger. Thank you very much for your suggestion.

[Comment 2] At the state-of-art section, there are some questions that should be review.  Figure 1 is only about children? Only about adults? Both? It should be detail each part. On the other hand, in the Electrophysiological evidence epigraph, when you are explaining the importance of stress and pain catastrophizing what are the clinical implications of this findings? Are there evidence about management of that? Also, on page 11, when you are exploring the importance of the emotional processing. What are the clinical implications?

Response: Figure 1 is a schematic of the central and peripheral mechanisms associated with pain chronification (as known from adult human and rodent studies). We overlayed Figure 1 with evidence from pediatric literature (which included rat pups and human pediatric studies) in blue ink to Figure 1. We have now added text to make this clear. Please see L.139-141, P. 3. And please see “The main adult mechanisms are mentioned in bold black color denoted by letters, with pediatric specific features that contribute to pain chronification overlaid in blue colored text. “ in the legend of Figure 1, L. 152-153, P.4.

Thank you for emphasizing the importance of providing clinical implications of ACEs and emotional aspects for management of pediatric chronic pain. We added references and provided evidence on this comment (P.14, L.627). We have specifically not elaborated on psychological mechanisms in this review as they have been elaborated on previously.

Understanding the role of developmental imaging studies and MRI biomarkers in diagnosis and treatment of pediatric pain is crucial for the development of personalized care. We added more evidence on the Default Mode Network (emotional processing) signatures of chronic pain in children. (P.8 and P11)

[Comment 3] In the future directions for clinical practice epigraph, in Table 1, which is very interesting you expose some mechanism for cortical modulation of pain, such us integrative care relaxation, which are the references?

Response

The references related to integrative care, musical therapy etc. have been added to the text within the manuscript. Please see the added text L. 825-827, P. 17 for the references outlining the utility of integrative care and music therapy approaches in the management of pediatric chronic pain.

[Comment 4] Conclusions were correct but should also include a summary of the clinical implications.

Response:

We have now added clinical implications to the conclusions section (L.847-873, P.19).